# Mind the Gap Between Conversations
# for Improved Long-Term Dialogue Generation

**Qiang Zhang, Jason Naradowsky, Yusuke Miyao**
Department of Computer Science
The University of Tokyo
{qiangzhang714, narad, yusuke}@is.s.u-tokyo.ac.jp

## Abstract

Knowing how to end and resume conversations over time is a natural part of communication, allowing for discussions to span weeks, months, or years. The duration of gaps between conversations dictates which topics are relevant and which questions to ask, and dialogue systems which do not explicitly model time may generate responses that are unnatural. In this work we explore the idea of making dialogue models aware of time, and present GapChat, a multi-session dialogue dataset in which the time between each session varies. While the dataset is constructed in real-time, progress on events in speakers' lives is simulated in order to create realistic dialogues occurring across a long timespan. We expose time information to the model and compare different representations of time and event progress. In human evaluation we show that time-aware models perform better in metrics that judge the relevance of the chosen topics and the information gained from the conversation.

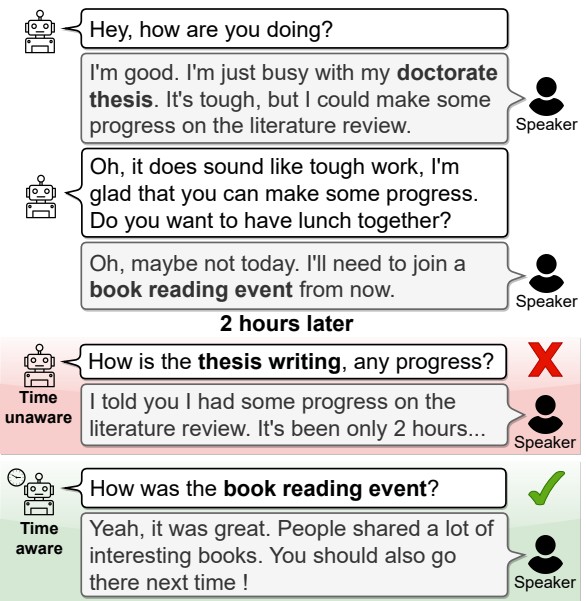

Figure 1: An illustration of the time aware dialogue models we propose. The time aware model talks about events that are informative and effective in a conversation considering the duration of the event and the gap between sessions.

## 1 Introduction

As language models scale to unprecedented sizes, so too has their ability to generate reasonable and fluent responses to human dialogue. However, one limitation of large language models (LLMs) towards creating human-like dialogue lies in the fundamental distinction that human cognition is *embodied*, while LLMs are not. Any considerations which stem from the human physical experience may not be well-represented explicitly in the training data dialogues of LLMs, as reiterating information known to both parties would violate the Gricean maxim of quantity (Grice, 1975), and may not be effectively utilized during generation.

In this paper we focus on one aspect of embodied experience – time – and the role that awareness of time plays in shaping realistic dialogue in multi-session conversations. We argue that an awareness of the passage of time between conversations allows human speakers to more accurately gauge which topics and questions will result in new information, leading to conversations which are more informative and appear more natural. For example (Figure 1), a dissertation is a time-consuming endeavor, so if given a two-hour gap between conversations, it is unlikely that significant progress has been made. In the context of potential *information gain*, asking for a progress update may be less useful than asking about an event with more expected progress within that time frame, such as a lunch or a meeting. To enable dialogue models to behave similarly, we propose incorporating an awareness of time as it pertains to (1) how much time has passed since the previous conversation, (2) what events were previously discussed, and (3) what progress of each event is expected over that duration.

Previous work introduced the concept of multi-session chat (MSC) and dataset for the task where the discussion between two speakers is divided with gaps between sessions (Xu et al., 2022a). However, gaps between sessions are relatively short (1-7 hours, or 1-7 days), and event topics are derived from basic persona attributes (*I have six cats*) which prevents any discussion of long-term events or event progress. Moreover, annotators vary between sessions, which may have further reduced consistency.

To remedy these issues and support long-term MSC research, we present GapChat, a dataset which extends the MSC dataset with additional sessions [1]. However, while chat in MSC is open-ended, conversations in GapChat are based on simulated timelines. This design choice allows for two participants to create a realistic and consistent conversation about long-term events, with gaps between sessions on the scale of days, weeks, or months, in a comparatively shorter period of time.

We explore multiple ways of making dialogue models time-aware, using information such as the ongoing events of speakers, the expected duration of the events, and the duration of time gap between sessions. Our contributions are as follows: (1) the creation of a new MSC-style dataset that simulates explicit gaps in communication, across various time-scales, to enable research into long-term MSC dialogue generation, (2) the exploration of different time-aware dialogue models, which incorporate time information into their contexts to enable varying degrees of time-based reasoning about discourse topics, and (3) we demonstrate via human evaluations of generated dialogues that the inclusion of time information improves the naturalness, informativeness, and relevance of conversations.

## 2 Related Work

**Pragmatics Theory of Communication** Underlying this work is the cooperative principle of communication (Grice, 1975), in which communication is goal-oriented, and where effective conversations are those which adhere to basic principles, including the desire to communicate only information which requires it. The cooperative principle has been used as a means to evaluate both human-human (Eskritt et al., 2008; Kleinke, 2010) and human-computer (Lan et al., 2020; Langevin

et al., 2021) conversations. To assess the quality of follow-up questions posed by artificial agents, Ge et al. (2022) proposed measures based on Gricean maxims to capture aspects such as relevance, informativeness, truthfulness, clarity, and coherence. Motivated by this line of work, we explore the role that time awareness plays in gaining information effectively by avoiding discussing non-mentionable topics.

**Long-term Dialogue Model** Research on long-term dialogue generation based on neural networks has primarily focused on means of storing and accessing long-term information beyond the bounds of a limited context window. One approach to this problem is selecting valuable information for long-term storage, such as conversation summaries (Xu et al., 2022a) and persona attributes (Xu et al., 2022b). This approach involves training a model which can retrieve from storage information which is relevant to the current topic. An alternative to retrieval, attention mechanisms can fulfill a similar role (Zhang et al., 2022).

**Dialogue Models on Topic Selection** A potential benefit of time-awareness is topic selection that results in more informative conversations. Previous approaches have used semantic relevance as the basis for topic selection (Somasundaran et al., 2020; Xu et al., 2021), or have selected based on topic transition patterns (Xie et al., 2021; Ling et al., 2021). While such factors are shown to be effective in general discourse, in the context of multi-session discourse with gaps between conversations, commonsense about previous topics may help rule out topics that are unlikely to result in new information being discussed. If this is a guiding principle in human discourse, conversations that utilize temporal information should appear more natural.

## 3 GapChat: A Time-Aware MSC Dataset

The MSC dataset contains gaps between sessions and their durations are annotated. However, the gaps were short, and the annotators were changed between sessions. They were not instructed to pay attention to the relation between topics, events and the duration between gaps. Thus temporal information in MSC has been shown to have a minimal effect on model performance (Xu et al., 2022a). However, we hypothesize that the use of a dataset with discussions focusing on more realistic long-term events may have a more significant impact

[1] https://github.com/QZx7/MindTheTime/tree/main

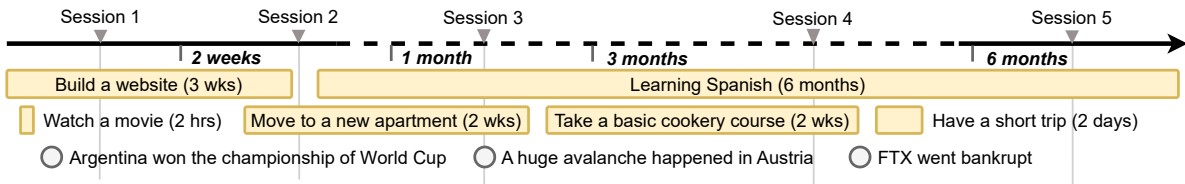

Figure 2: An example of the simulated timelines for a speaker in GapChat. Each speaker will engage in specific events that take a certain period of time to finish. Yellow rectangles and circles indicate life events and grey circles indicate world events.

on model performance. To test this hypothesis, we create a new conversation dataset, GapChat. We follow the general settings in MSC and similarly collect dialogue data via crowdsourcing. Thus GapChat can also be used as additional data for any task utilizing MSC. However, unlike MSC, conversations in GapChat are not completely open-ended, but are grounded to the hypothetical lives of the two annotators, who are each given a procedurally generated timeline of events in their lives. This notion of a simulated life provides speakers with updates to events in their lives which are realistic with respect to the time which is said to have passed since the last chat session. Sessions are then scheduled to take place randomly at various positions along the timeline, which may occur before, during, or after any particular event.

### 3.1 Events and Timelines

We define a *timeline* as a linear sequence of (possibly overlapping) events, where each event is represented by a string label (*watch a movie*) together with an expected duration (*2 hours*). When two annotators are selected, a series of events is sampled, and timelines created for both speakers by randomly ordering the sampled events. There are two types of events: life events and world events.

**Life Events**   A life event is an event that is said to happen in the person's life and they are exclusive to each speaker. We use life events to ensure the continuity in the multi-session conversations. Life events are crafted in two ways: manually, and generated with ChatGPT [2]. When crafting events manually, we collect example life events from online resources [3]. The durations of these events are estimated by searching online forums (e.g., Quora,

Reddit) using the query "How long will it usually take to finish <event>?". Up to 5 answers are selected, and the average estimation is used as the duration.

To include a more diverse range of events in our dataset, we also use ChatGPT to generate additional life events that vary in duration and domain. When generating with ChatGPT, we prompt the model to "Generate a list of events or daily activities that require around <duration> to finish," where <duration> represents the time needed to complete the event (e.g., 1-3 weeks). Manually and generated, a total of 50 life events with varying durations, including hours, days, weeks, months, and years are collected, with 10 events for each duration.

In order to reflect changes in the topic over time, longer events are further subdivided into a series of steps denoted as an event schedule. To do that, we ask ChatGPT to generate the steps towards finishing each event via prompting (see Figure 11). For instance, in the case of attending a 3-day basic cookery course, the introduction may require 1 day to complete, learning knife skills may take another day, and learning to cook basic dishes may take 2 more days. And these three steps make up a schedule for the event of "attending a basic cookery course". A dialogue session could randomly be assigned to occur at any place within this event, or not at all. We create event schedules for each event that is with a duration that is longer than "1 hour". Each schedule of an event consists of a maximum of 7 steps. For events longer than a month, we generate two schedules with varying sets of steps, each representing a different approach to completing these long-term events. Examples of life events and their corresponding schedules can be found in Appendix A.

**World Events**   Not all discussion topics pertain to the lives of the speakers, and it is common to also have discussions on current events. To account for

[2] https://platform.openai.com/docs/models/gpt-3-5
[3] https://simplicable.com/philosophy/life-events; https://www.dudleycourtpress.com/50-life-events-for-your-own-memoir/

| #Session | Dialogues | Utterances |
|----------|-----------|------------|
| 3        | 150       | 8,235      |
| 4        | 300       | 26,651     |
| 5        | 200       | 21,368     |
| Total    | 650       | 56,254     |

Table 1: Data statistics of GapChat.

this and to add further realism to the conversations, we also include world events. We define a world event as a newsworthy event which takes place in the world, and is thus contained in the timelines of both speakers, such as "*Argentina wins the World Cup championship*". We collected 78 world events by extracting news article titles from Google News between November 2022 and January 2023. When used in timelines, world events happen in the same order as in the real world.

### 3.2 Data Collection

We collect data via Amazon Mechanical Turk (AMT), beginning with the easier task of 3-session conversations and inviting reliable workers to take part in the collection of 4 and 5 session conversations. A total of 48 workers were selected to participate in the final data collection. We require fluency in English for all participants. Workers are paid $7 per hour on average. The instructions and interface used for data collection can be found in Appendix C.

**Initial session**  At the beginning of the initial session, each speaker is provided the first event in their timelines (e.g., "You just started attending a basic cookery course, which would take about 4 days") as an initial event to share in the conversation. The initial events are randomly selected from the life events. Once the minimum session length is reached, either speaker can end the current session, move time forward, and create a new session with a randomly generated time gap ranging from 10 minutes to 1 year.

**Subsequent sessions**  For all subsequent sessions, speakers are provided with updates on the progress of their ongoing events based on the duration, steps of the events, and the time gap. If the time gap is shorter than the minimum required time to reach the next step in the schedule of the event, speakers will receive a message stating "No significant progress." Conversely, if the time gap covers the

> In the following conversation, speaker_1 and speaker_2 are updating their daily lives. In the conversation, both speakers might have mentioned some events. The events might have been finished or are currently going on and just started. Extract only the events that the speakers are currently going on and just started with the following conditions.
> **#Conditions:**
> 1. Summarize the events as nouns or noun phrases, such as "going for a tirp", "starting a MBA program", "taking an online course", "building a swimming pool".
> 2. Describe the events as brief as possible using the shortest summary.
> 3. Generate the answers in the format of "speaker_1 : <event_1>, <event_2>.\nspeaker_2: <event_1>, <event_2>".
> 4. "speaker_1" and "speaker_2" must be in lowercase, and there must be a "\n" before "speaker_2".
> 5. If the speaker did not mention any events, generate: " <speaker>: Not mentioned."
> **#Conversation:**
> {speaker_1: Hi, how are you doing?
> speaker_2: I'm doing great. How about you?
> ...}

Figure 3: The prompt we use to extract events from a given conversation during inference.

full duration of the event, speakers will be notified that the event has been completed and new life events will be provided according to the timeline.

**Session details & statistics**  A typical MSC session contains 12-14 utterances, and this may make it difficult for speakers to engage in in-depth discussions before transitioning to a new session. We observe that many sessions in MSC immediately continue the conversation of the previous session, essentially making the data no different than a longer single session. To address this, we design each session in GapChat to have at least 20 utterances and to start and end in a natural way (e.g., greetings and closures), and we encourage sessions to continue past this minimum. Additionally, we instruct speakers to be mindful of the time gap between sessions as it may influence their behavior in the subsequent session. Table 1 shows the statistics of GapChat. In total we collect 56,254 utterances across 2,650 sessions. Each conversation consists of 3∼5 sessions. Examples of a 4-session conversation is available in Appendix B. For evaluation we split the data to Train/Valid/Test with a ratio of 0.7/0.1/0.2.

## 4 Modeling Time

We propose a method of modeling time in multi-session dialogue systems. As shown in Figure 4, this consists of three procedures: 1) extracting on-

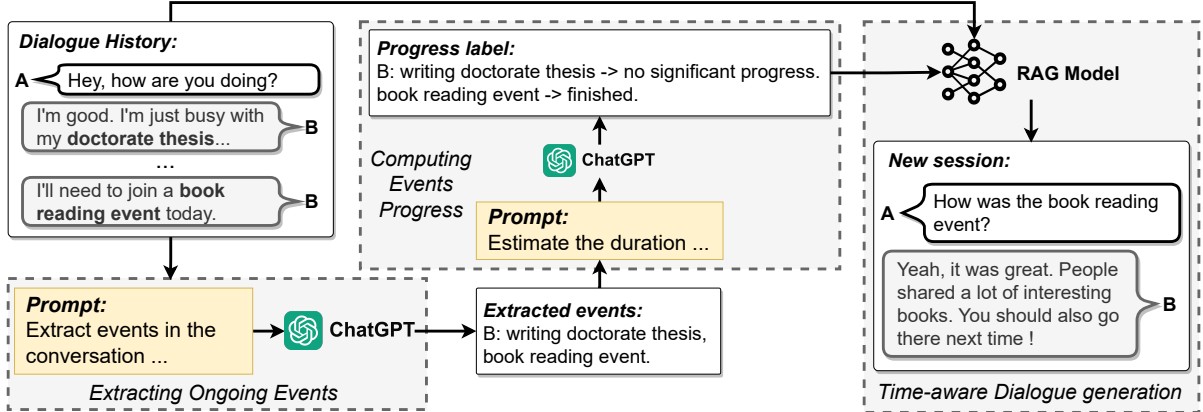

Figure 4: The illustration of the process of modeling time with progress label. When using schedules, an additional step of prompting LLM for schedule information is added.

going events, 2) computing events progress, and 3) time-aware dialogue generation.

## 4.1 Extracting Ongoing Events

At training time, dialogue events are observed, since they are pre-defined in timelines that are assigned to the speakers. However, at inference time, it is necessary to extract ongoing events on-the-fly from the dialogue history. We phrase this as a text generation task where the input is the preceding dialogue history and the output is a list of events discussed in the history. The events to extract are defined as in Section 3.1. For instance, if a speaker mentioned "I'm currently working on a research project until the end of the semester", the event "working on a research project" is extracted as a life event. We extract events in a prompt-based manner using ChatGPT. Different styles of prompts are explored, as shown in Table 12~ 14 in Appendix D. The prompt which resulted in the highest extraction performance is shown in Figure 3, and is used in the following experiments.

## 4.2 Computing Events Progress

During training, we compare the event duration against the session gap to determine the progress of events. At inference time, we first estimate the duration of the events extracted in the previous procedure through prompting. We use two styles of methods to represent the progress of the events: the *progress labels* and *event schedules*.

**Estimating event duration**   Event duration is estimated by querying a knowledge base that has the temporal commonsense knowledge as in the MC-TACO dataset (Zhou et al., 2019). We utilize Chat-

Given a list of events provided by two people, speaker_1 and speaker_2, estimate a typical time duration to finish each event. For each event, label it with a time duration tag in the following steps.
**#Steps:**
1. select a base time range from {hour, day, week, month, year} using the commonsense knowledge.
2. select a number that is associated with the base time range to form the final time duration tag.
3. generate the answer with <speaker_id>: <event> -> <number><base time range>.
4. use N/A if the events for a speaker are not provided. The generated text contains only the answer.
**# Example:**
List of events:{"speaker_1": [" just started a one-year collage program."],"speaker_2": [" getting driving license"]}
Answer: speaker_1: just started a one-year collage program. -> 1 year\nspeaker_2: getting driving license -> 2 months
**#Question:**
List of events:
{"speaker_1": ...}

Figure 5: The prompt used to estimate event durations.

GPT as a knowledge base, and prompt it to provide a typical duration for each event. Figure 5 shows the prompt we use for estimating event duration.

**Using progress labels**   When using progress labels to represent the event progress, we formulate event progress as a discrete labeling task, where each extracted event in the previous procedure is labelled with one of five progress labels, {*No significant progress, 1/4 finished, half finished, 3/4 finished, finished*}. Finer or real-valued estimation of progress is possible, but was not pursued due to potential sparsity issues. Event durations are then compared with the time gap to calculate the progress labels for each event. For instance if the

duration of the event "getting a driver license" is "2 months" and the time gap is "6 weeks", the event is given a progress label of "3/4 finished". Although it is possible to prompt ChatGPT directly to generate the progress labels, we calculate these labels to prevent potential inaccuracies or hallucinations that ChatGPT might produce regarding mathematical tasks (Bang et al., 2023).

**Using event schedules** As an alternative to the more numerically oriented progress labels, the schedules collected in Section 3.1 contain steps that are required to finish an event, and thus can also represent the progress of the events. We leverage the schedules and split a schedule into two lists "finished" and "to-do". The "finished" part contains those steps that can be completed during the session gap, and the "to-do" part contains the remaining steps. For instance, a schedule of the "getting the driver license" contains 5 steps of "one week for learning rules, 2 weeks for practicing, 2 weeks for passing exams, one week for road check, one week for getting license." If the time gap is "two weeks," the "finished" list includes "one week for learning rules," and the remaining steps are added to the "to-do" list. The "finished" part of the schedule represents the current progress of the event.

### 4.3 Time-aware Dialogue Generation

We use a RAG (Retrieval-Augmented Generation) 2.7B model (Lewis et al., 2020) for dialogue generation. A RAG model utilizes a retriever to retrieve related contexts stored as documents in memory, making it effective at handling large collections of text. The truncation of this model is set to 1024, enabling it to encode more context.

**Documents and retriever** Following MSC, we save the dialogue context as documents and retrieve with a DPR model (Karpukhin et al., 2020). Different sessions are saved as separate documents in the memories encoded by the DPR model. When provided with the dialogue context, the top-5 documents are retrieved for response generation.

**Training the model** During training time, events and different types of time-aware information (progress labels and schedules) are combined with dialogue history as context in a manner similar to Persona-Chat (Zhang et al., 2018) (Table 2).

We train different time-aware RAG models (TA-RAG) by providing various time-aware information (gaps, progress labels and schedules) using the

---

A: Hey, how are you doing?
B: I'm good. I'm just busy with my doctorate thesis.
...
B: I'll need to join a book reading event today.
**Events**
B: writing doctorate thesis, book reading event.
**Progress**
B: writing doctorate thesis [no significant progress], book reading event [finished].
**Gap**
2 hours

---

Table 2: The sample input to time-aware models.

MSC-RAG 2.7B 1024 as the initial model. A baseline model, RAG (FT), is also trained with only the dialogue history and extracted events but without time-aware information. All models are trained on the ParlAI platform [4] in a seq2seq style, with the dialogue context as the input and the response the label. The models are trained with 2 NVIDIA A100 GPUs for 72 hours.

## 5 Experiments

Models are separated into two groups, time-unaware models and time-aware models. Human evaluation is conducted to compare all models against RAG (FT).

### 5.1 Time-unaware Models

We include comparisons to a number of models which do not explicitly consider time gap and event duration information as baselines:

**MSC-RAG** The RAG 2.7B model proposed by MSC (Xu et al., 2022a), where the model saves previous dialogue sessions and retrieves relevant information during generation.

**RAG (FT)** MSC-RAG model fine-tuned on GapChat. No time-aware information is provided in this model.

### 5.2 Time-aware Models

**TA-RAG** Time-aware models are based on the MSC-RAG model, fine-tuned on GapChat, and using various time-aware information described in Section 4. For the TA-RAG (progress) model, the

---

[4]https://parl.ai/

| | Model | Human Ratings against RAG (FT) | | | | |
|---|---|---|---|---|---|---|
| | | Naturalness | Informativeness | Relevance | Time-Awareness | Total |
| **Time-** | MSC-RAG | - 3.17 | 1.12 | - 4.68 | - 44.10 | - 12.71 |
| **unaware** | ChatGPT | 7.64 | - 0.50 | 1.00 | 20.28 | 7.11 |
| | TA-RAG: | | | | | |
| | *progress* | 9.02 | 5.62 | 13.75 | 44.82 | 18.30 |
| | *schedule* | 9.16 | 4.06 | 15.22 | **54.84** | 20.82 |
| **Time-** | *both* | 15.15 | **6.32** | **20.78** | 50.56 | **23.20** |
| **aware** | ChatGPT: | | | | | |
| | *gap only* | 10.52 | 2.18 | 3.12 | 16.66 | 8.12 |
| | *progress* | 18.08 | 5.12 | 15.38 | 38.78 | 19.34 |
| | *schedule* | 15.84 | 4.54 | 12.30 | 47.78 | 20.12 |
| | *both* | **18.26** | 3.80 | 17.30 | 52.58 | 22.99 |

Table 3: Human evaluation results of different models when compared against RAG (FT). Negative numbers indicate that the model performs worse than RAG (FT).

time-aware information is provided in the style of progress labels, and for the TA-RAG (schedule) model, it is in the style of event schedules. TA-RAG (both) indicates that both styles of time-aware information are provided.

### 5.3 ChatGPT

Besides the fine-tuned models, we also add Chat-GPT into our experiments to explore its time-awareness. We consider both time-unaware and time-aware scenarios and prompt ChatGPT to generate responses. In the time-unaware case, only dialogue history and extracted events are provided to the model. In the time-aware case, we explore different types of ChatGPT settings. In the "with gap only" type, in addition to the dialogue history and events, time gaps between sessions are explicitly given, enabling ChatGPT to recognize the engage in a multi-session conversation. In the "progress", "schedule" and "both" types, progress of events are also provided as in TA-RAG models.

### 5.4 Evaluation

**Collecting dialogues**   Models are evaluated using the human evaluation by comparing the conversations they generate. Conversations from all RAG models are generated in a self-chat style, where the RAG models engage in a conversation with a common BlenderBot3B model. Self-chat has been shown to perform comparably to human-model conversations (Smith et al., 2022) for evaluation purposes. In the case of ChatGPT, we utilize prompts to obtain responses (Figure 13, Appendix E).

Conversations are generated session by session, where the first 3 utterances of the first session are seeded from manually crafted scripts to ensure the models start the conversation by sharing some events. In subsequent sessions, randomly selected time gaps and new events are provided together with the dialogue history of the previous session to the models. All models use the same time gaps and events to ensure a consistent experiment setting.

Evaluation follows ACUTE-Eval (Li et al., 2019), where annotators are asked to rate the conversations generated by various models in comparison to the baselines. In our experiments, we compare all the models to RAG (FT) and evaluate the conversations session by session. The annotators are provided the dialogue history of previous sessions, the event information and the time gap. They are then asked to select the conversations generated by one model over another. For each model, we collect 150 multi-session conversations spanning different session lengths (50 for 3-session, 4-session, and 5-session scenarios, respectively).

**Human evaluation questionnaire**   In the evaluation, annotators are asked a total of 11 questions, grouped as 4 attributes: naturalness, informativeness, relevance and time-awareness. Naturalness refers to the ability of the model to generate conversations that feel like two friends updating each other on their daily lives. Informativeness evaluates whether the model frequently asks frustrating questions about event progress, affecting the information gain (Ge et al., 2022).

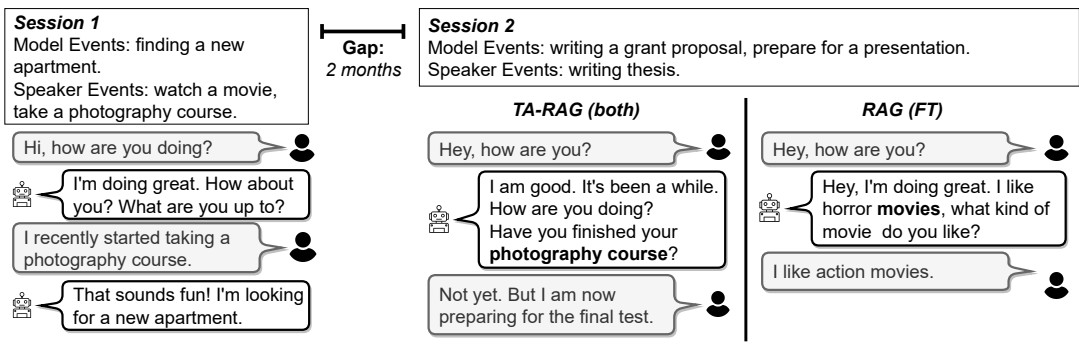

Figure 6: A sample conversation generated by RAG (both) and RAG (FT). The models generate the conversations with the same events and time gap.

Annotators are asked, "Which speaker asks annoying questions about events with no significant progress?" Relevance measures the model's ability to generate follow-up sessions related to mentioned events by asking annotators "In which dialogue does the speaker ask/talk more about relevant events?" Time-awareness assesses the model's ability to identify time gaps and the progress of events by asking annotators "In which dialogue does the speaker identify time gaps more accurately?"

We conduct the evaluation on AMT. Questions are phrased differently for each attribute and average ratings are calculated (Appendix F). After pilot tests, 66 annotators are chosen for the final evaluation task. 10 annotators are hired for each task of model comparison. We also request explanations for choices to validate responses. During analysis, we filter out evaluations with short working time (< 200 seconds) and unreasonable explanations (e.g., single-word, repetitive, unrelated to the conversation, or copied from other text). The Fleiss' Kappa across all annotators is $K = 73.1\%$.

## 6  Results & Analysis

### 6.1  Main Findings

Results of the human evaluation are shown in Table 3. Our main finding is that **time information improves overall naturalness of the conversations**, which supports the hypothesis that time-based reasoning of discourse topics is an important part of long-term multi-session dialogues. Figure 6 shows an example in which TA-RAG (both) is able to produce more natural conversations (more examples in Appendix G). TA-RAG with one type of time information performs worse than ChatGPT on naturalness, however, overperforms ChatGPT when both types of time information are added.

Between different types of time information, we find that **progress labels contribute most to the enhancement of informativeness, but schedule information has relatively minor impact on the informativeness.** This may be attributable to the more direct manner in which this approach provides information to the model, as the progress towards an event represents the outcome of a time-based reasoning process that is performed outside of the model. Using the progress representation would closely align with a desired discourse action (e.g., do not select events with labels of "no significant progress"), whereas the schedule representation requires the model to perform additional reasoning.

### 6.2  Analysis

**Various gap duration**  We analyze the performance of the models with regard to specific gap durations, but do not find a consistent relationship between gap length and the evaluation measures (Appendix H). In some models (TA-RAG (both)) we find that naturalness and time-awareness are more stable over different session gaps, whereas informativeness and relevance shows larger deviations (see Figure 7), but these deviations do not change the overall ranking of systems on each attribute.

**Time-awareness in ChatGPT**  ChatGPT is considerably larger and trained on more data than any other model in our study, and may have induced a better understanding of time. In fact, we rely on ChatGPT as a source of temporal commonsense when obtaining event durations. It is then no surprise that ChatGPT exhibits a certain level of time-awareness, however it does not score as high on this metric as TA-RAG models. Surprisingly,

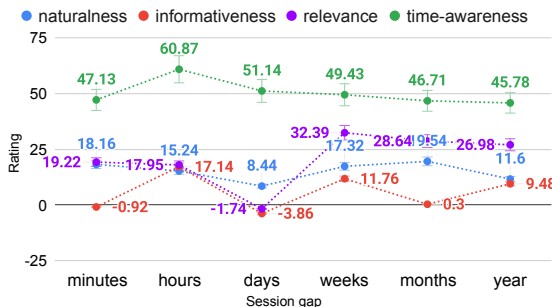

Figure 7: Performance of TA-RAG (both) over different session gaps.

| Model | #Correct Events |
|---|---|
| RAG (FT) | 17 |
| TA-RAG (progress) | 46 |
| TA-RAG (schedule) | 42 |
| TA-RAG (both) | 44 |

Table 4: Number of session-pairs in which the model selects correct events in a subsequent session. Total number of tested case is 60.

we found that while adding information about the gap between sessions improves naturalness, informativeness and relevance, it does not enhance the time-awareness of ChatGPT. Rather, we observed improved greetings and conclusions, and the inclusion of gaps appears to have primarily improved the overall structure of the dialogue.

**ChatGPT outputs lengthy responses resulting in relatively low informativeness.** To explore the impact of time information on ChatGPT's performance, we use similar prompts, as depicted in Figure 13, albeit with time information incorporated into the prompts. The inclusion of time information improves time-awareness, demonstrating that this information is useful and not implicitly utilized to the same extent by the original model. In terms of informativeness, we observe that adding both types of information results in lower performance than only utilizing progress labels. We find that this is due to the longer length of responses, which annotators naturally consider less informative. When schedule information is added, ChatGPT exhibits a tendency to incorporate schedule details into the reply, which increases the reply length.

**Adding time information helps the model select events in subsequent sessions.** We conducted an analysis of the generated conversations to examine the extent to which the TA-RAG models effectively select appropriate events as topics in subsequent sessions. We manually review 60 randomly selected consecutive session-pairs (20 for each of 3-session, 4-session and 5-session conversations) and measured the quality of topic selection as the extent the subsequent session avoided addressing events that have a label of "no significant progress". Table 4 shows the results of the conversations generated by different models. We observe that adding time information, especially the progress label in-

formation, results in selection of more appropriate events in subsequent sessions (where the more desirable events are ones which are most likely to cause discussion of new information).

## 7 Conclusion

This work is a study of the role that temporal reasoning plays in shaping dialogues over multiple sessions, and the extent to which temporal information is useful when integrated into existing dialogue systems. We demonstrate via human evaluations that time-aware models generate text with improved naturalness, informativeness, and relevance, resulting better topic selection and improved text generation quality overall. The results emphasize the importance for future dialogue models to consider "time-awareness" as an important factor in achieving natural conversations. Additionally, our analysis shows that incorporating time-aware information also enhances the performance of existing LLMs such as ChatGPT.

## Limitations

We rely on existing LLMs (ChatGPT) to serve as knowledgebases for temporal commonsense. The extracted event durations agreed with author intuitions, but as they are data-driven they may not reflect reality and may give a false expectation of the time necessary to complete events. The events used in our dataset represent a small set of possible events, and we do not know what coverage this system would have on events in real conversations. Further, using this system requires ChatGPT in the loop to extract events during inference time, and while accuracy was high and the extracted text was sensible on the events used in this study, there are inherent risks present when using an LLM for text generation, and incorporating that information downstream without human oversight.

## Acknowledgements

This work was partially supported by JSPS KAK-ENHI Grant Number 19H05692 and Moonshot R&D Program Grant Number JPMJMS2237_4.

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

## A  Samples of events in simulated timelines

Table 5 and 6 are examples of the life events. An event can have different schedules that lead to different consequences/ends. We use these schedules to increase the diversity of the directions of developing the ongoing events. For each life event, we pre-define the duration and use ChatGPT to generate steps of a schedule for the event within the duration via prompts. An example prompt can be found in Figure 12. Table 7 shows an example of the world event.

## B  Samples of GapChat

Table 8∼11 show a 4-session sample conversation we collect. For each session in the dataset, we provide the progress of events and the utterances. For subsequent sessions, we also provide the session gap since the previous session.

## C  Instruction and interface for data collection

Figure 8 and 9 show the instructions we use for data collection. We explicitly ask speakers to follow the events and timelines in the instruction. We also raise the point that any offensive, abusive content will not be allowed and the speakers are able to report and stop and task any time. We also provide positive and negative examples to demonstrate how the data should look like for the speakers.

To generate the conversations, we use a matching system to first randomly match two participants. After matching, the participants are redirected to the chat room as shown in Figure 10. At the beginning of the conversation, each participant will be shown their initial events for the participants to talk about. When the session reaches the maximum length, either participant could move the time forward for a random time gap by clicking the "End current session" button. And for the new session, we show different types of information to both participants. Finished progress indicates the life events they were engaging in the previous session. The progress is represented as the schedule steps based on the time gap. We also show some random life events and world events defined in the timeline. Finally, we also provide some future plans for the participants. The future plans are either some future schedule steps for unfinished progress or new events in the timeline if the previous events are finished.

## D  Prompts for modeling time

### D.1  Prompts we used in our experiment settings

Figure 11 is the prompt we use to generate the steps (schedule) towards finishing a life event in our training data.

Figure 12 is the prompt we use to generate short schedules for given events. These prompts are used as they show best performance in our tests.

### D.2  Prompts we tried with different styles

We try different types of prompts following previous works researching on the factors that could affect the performance of prompting results (Schick and Schütze, 2021; Lu et al., 2022; Mishra et al., 2022). Only few-shot prompting method is explored because we require the generated events to be in certain format (Table 12∼14).

For the question-answering style, we only fix the first two questions and dynamically select the questions based on the answer to previous question. For instance, if the answer to Question 1 is "Yes", we will choose "What are the events that speaker A is engaging?" as Question 3. When asking a question, all the question and answers to previous questions are included as part of the prompt.

| Life event |
| --- |
| You just started preparing and executing a social media marketing campaign for your company, which would take about 3 months. |
| *Duration*: 3 months |
| *Schedule* |
| **Steps:**
4 weeks for researching the audience and markets, one week for creating engaging content that aligns with the campaign goals, one week for designing and setting up the campaign, 4 weeks for executing and optimizing the campaign, one week for analyzing the campaign data, one week for making adjustments to the content and strategies as necessary. |

Table 5: An example of the schedules in simulated timelines.

| Life event |
| --- |
| You just started writing your doctor thesis, which would take about one year. |
| *Duration*: 1 year |
| *Schedule 1* |
| **Steps:**
one month for reviewing the guidelines and outlining the structure of the thesis, 2 months for writing the introduction, 2 months for writing the remaining chapters, 2 months for revising the thesis based on feedback from other colleagues, one month for preparing the formatting and citations, one month for final tough and submitting to committee, one month for addressing comments requested by the committee members and defending the thesis. |
| *Schedule 2* |
| **Steps:**
one month for reviewing the guidelines and outlining the structure of the thesis, 2 months for writing the introduction, 2 months for writing the remaining chapters, 2 months for revising the whole thesis for clarity, coherence and flow, one month for revising the thesis based on feedback, 2 months for preparing and submitting a conference paper, one month for preparing the extension procedures of the doctorate program. |

Table 6: An example of multiple schedules that lead to different results of the same life event.

| World events |
| --- |
| Argentina won the championship of World Cup.
Prince Harry reveals whether he's circumcised in bombshell 'Spare' memoir
US closes in on Bankman-Fried Inner Circle with probe of FTX chief engineer. |

Table 7: Examples of world events.

*Session 1*

Events:

Speaker 1: You just started organizing the storage room, which would take about 2 days.

Speaker 2: You are about to have a short vacation to the beach, which would take about 3 days.

---

Speaker 1: Hi! How are you doing? I heard you're going on a short vacation to the beach. That sounds exciting!

Speaker 2: Hey! Yes, I'm really looking forward to it. It's been a while since I've had a break.

Speaker 1: Definitely, it's always nice to take some time off. Do you have any plans for what you'll do while you're there?

Speaker 2: Not really, just planning to relax and soak up some sun. Maybe take a dip in the ocean.

Speaker 1: That sounds like the perfect way to unwind. Have you packed everything you need for the trip?

Speaker 2: Yes, I have everything sorted. I just need to double-check that I haven't forgotten anything important.

Speaker 1: That's great to hear. On my end, I've been organizing my storage room. It's been a bit of a mess, so I've been taking some time to sort through everything.

Speaker 2: Oh wow, that sounds like quite the task. How's it going so far?

Speaker 1: It's definitely been time-consuming, but it's starting to look much better. I've been able to clear out a lot of clutter and create more space.

Speaker 2: That's great! It always feels good to have a more organized living space.

Speaker 1: Absolutely. Speaking of which, have you been keeping up with your own home organization?

Speaker 2: Yeah, I've been trying to keep things tidy. It's a never-ending process though!

Speaker 1: Tell me about it. It always feels like there's something to clean or organize. Any other plans for when you return from your vacation?

Speaker 2: Not really, just getting back into the routine of things. How about you?

Speaker 1: I'm planning to work on some writing projects when I finish organizing the storage room. I've been meaning to get back into it for a while.

Speaker 2: Oh, that's interesting! What kind of writing do you do?

Speaker 1: Mostly fiction, but I've been wanting to try my hand at some non-fiction pieces as well.

Speaker 2: That sounds like a lot of fun. Let me know if you need any help or feedback.

Speaker 1: Thanks, I appreciate it! How about we check back in after your vacation and see how everything's been going?

Speaker 2: Sounds good to me! Have a good one.

Speaker 1: You too, enjoy the beach! Talk to you soon.

Speaker 2: Thank you..

---

Table 8: Session 1 of an sample conversation we collect.

*Session 2*
Gap: 1 week.
Events:
Speaker 1: You just started to prepare for a marathon context, which would take about one year.
Speaker 2: You are about to visit the nearby town for about two days.

---

Speaker 1: Hey! It's been a week since we last talked. How have you been?
Speaker 2: Oh, hi! I've been great, thanks. How about you?
Speaker 1: I'm doing well, thanks for asking. By the way, how was your beach vacation?
Speaker 2: It was amazing! The beach was so beautiful and the weather was perfect. I even got a nice tan.
Speaker 1: That's awesome to hear! So, I finished organizing the storage room, and now I've started preparing for a marathon contest that will take about a year. What about you? Did you have any progress in your recent events?
Speaker 2: That sounds interesting! Well, I've been studying for a certification exam, and I also planned and executed a fundraising campaign for my local community center.
Speaker 1: Wow, you've been busy! How did the fundraising campaign go?
Speaker 2: It went really well! We managed to raise a good amount of money, and the community center was very grateful for our help.
Speaker 1: That's great to hear!
Speaker 2: Same great to hear!
Speaker 1: By the way, have you heard about ChatGPT's AI making puzzles that will make you want to throw brickbats?
Speaker 2: No, I haven't.
Speaker 1: Well, apparently they're really challenging and frustrating, but also addictive.
Speaker 2: What are these puzzles about?
Speaker 1: Some people are saying they can't stop playing them.
Speaker 2: Oh, I see. That sounds like fun, but also kind of frustrating. Anyway, what are your future plans for the next 6 months?
Speaker 1: I'm planning to work on a personal project and pursue a passion that I've been neglecting for a while. How about you?
Speaker 2: I'm planning to spend Monday and Tuesday working on a presentation that I have to give on Wednesday.
Speaker 1: Good luck with that!
Speaker 2: It's going to be a busy couple of days, but I'm looking forward to it.
Speaker 1: Let me know how it goes.

Table 9: Session 2 of an sample conversation we collect.

*Session 3*
Gap: 18 hours.
Events:
Speaker 1: You watch TV dramas.
You play video games.
Speaker 2: Tour the town's main attractions.
Have lunch at a local restaurant.

---

Speaker 1: Hey, how's it going?
Speaker 2: I've been busy with a lot of things
Speaker 1: It's been almost a day since we last talked. What have you been up to?
Speaker 2: Nothing really exciting to talk about. How about you?
Speaker 1: Well, I've been working on my storage room organization project.
Speaker 2: That's great!
Speaker 1: I finished it yesterday, and now I'm looking forward to starting something new.
Speaker 2: Congratulations on finishing it. So, what's your next project going to be?
Speaker 1: I'm planning to participate in a marathon that's going to happen next year, so I just started preparing for it.
Speaker 2: Wow, that sounds like a big challenge.
Speaker 1: Yes, my goal is to complete it within a specific time frame.
Speaker 2: Do you have any specific goals for the marathon?
Speaker 1: I'm also planning to set smaller goals for myself, like improving my running time and endurance.
Speaker 2: That's impressive. Good luck with your training. By the way, did you hear about the latest news on Microsoft and Activision?
Speaker 1: No, what happened?
Speaker 2: Apparently, they backed off from their aggressive claim in the FTC case. It's quite interesting to see how these big companies handle legal issues.
Speaker 1: Hmm, that's interesting indeed. I haven't been keeping up with the news much lately, but I did watch some TV dramas and play some video games during my free time.
Speaker 2: That's cool. I also spent some time assembling furniture that I bought last week. It was quite challenging, but I'm happy with the end result.
Speaker 1: Oh, nice! What kind of furniture did you assemble?
Speaker 2: It was a bookshelf and a cabinet for my study room. It took me a while to figure out the instructions, but I managed to get it done.
Speaker 1: Good for you! By the way, I'm planning to set aside some time to review and reflect on my personal and professional goals for the year and make any necessary adjustments. Do you have any plans for the next few days?
Speaker 2: Yes, I have to work on a proposal that's due on Friday, so I'm going to spend Wednesday and Thursday on it. How about you?
Speaker 1: I'm planning to choose the marathon and make a commitment to participate. I'm also planning to set realistic goals for myself and develop a plan to achieve them.
Speaker 2: Sounds like you're making progress. Keep up the good work!

Table 10: Session 3 of an sample conversation we collect.

*Session 4*
Gap: 3 hours.
Events:
Speaker 1: You do some housecleaning.
You play with your pet.
Speaker 2: You take a bath.
You do a board game with your family.

---

Speaker 1: Hey, it's good to talk to you again!
Speaker 2: Hey! It's been an interesting day.
Speaker 1: How have you been spending your time since we last talked?
Speaker 2: I've been watching some TV dramas, playing some video games, and assembling some furniture. What about you?
Speaker 1: Sounds like a nice day. I've been working on my presentation, and it's going pretty well so far.
Speaker 2: Thanks for letting me know. On my end, I'm planning to set aside some time to review and reflect on my personal and professional goals for the year. Do you have any tips on how to do that effectively?
Speaker 1: That's a great idea! One tip I have is to make sure you set aside enough time for reflection, and really focus on what's most important to you. You might also want to consider setting some specific, measurable goals to help you achieve your overall objectives.
Speaker 2: That's a good point. I'm also planning to choose a marathon and make a commitment to participate, as well as set some realistic goals for myself and develop a plan to achieve those goals. Do you have any experience with marathon training?
Speaker 1: Not personally, but I have friends who have trained for marathons before. It can be a challenging but rewarding experience.
Speaker 2: I'm excited to take on the challenge! By the way, I also wrote a short story this week. Would you like to hear about it?
Speaker 1: Of course! I'm always interested in hearing about your creative projects.
Speaker 2: It's a suspenseful story about a woman who becomes trapped in an elevator with a stranger who may or may not be dangerous.
Speaker 1: That sounds really interesting!
Speaker 2: I had a lot of fun writing it.
Speaker 1: Do you plan on doing anything with the story, like submitting it for publication or sharing it with friends?
Speaker 2: I'm considering submitting it to some literary magazines, but I haven't decided yet.
Speaker 1: That sounds like a good plan.
Speaker 2: I might also share it with some friends to get their feedback.
Speaker 1: I'm sure it will be well-received.

Table 11: Session 4 of an sample conversation we collect.

# Task Description

In this task, you have a natural conversation about specific events.

The events are some activities/events you are engaging in at the current time. (e.g., writing a novel, having a trip)

The conversation is multi-session style. And at the beginning of the first session, you will be assigned an initial event.

In the subsequent sessions, you will also receive other events and you need to talk about the progress of these events.

**Due to the various time duration between the sessions, some events might have significant progress to share, and some may not.**

**Be time-aware and judge what to talk about with your social skills and knowledge while paying close attention to the passage of time.**

Talk in a **natural** way as if you are having a conversation with your friends.

Do NOT talk about details about this task (e.g., This HIT is interesting!)

**Please make sure to check the negative examples.**

## Instructions

### Step 1: Match

When you are ready for the task, hit the **Start Matching** button at the bottom of this page.

### Step 2: Confirm your initial progress events

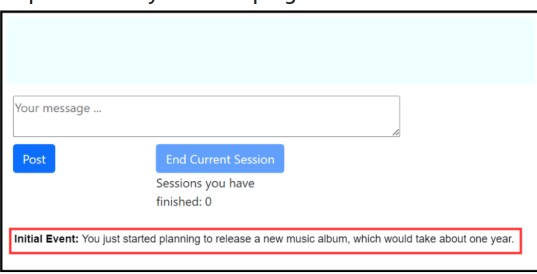

Before chatting, you will need to confirm your **initial event**.

E.g., *You just started writing your doctor thesis, which would take about one year.*

In the conversation, you can naturally share this event with another participant.

E.g., *Oh, by the way, I'm planning to release a new music album right now.*

### Step 3: Have the conversation for this session

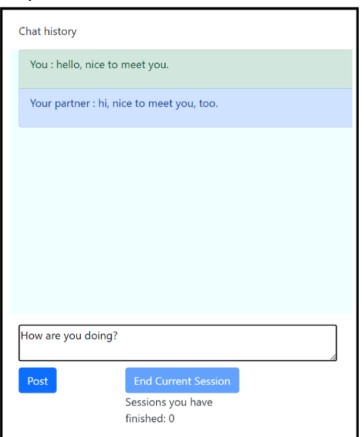

Input your text in the textarea and hit the Post button to send your message.

You can check the history any time during the task.

*Your messages can NOT be deleted once sent.* You can talk freely to make the conversation natural.

However, remember that you will need to

**1. ask progress of your partner's events.**

**2. share the progress of your events if you're asked.**

### Step 4: End current session and move the time forward

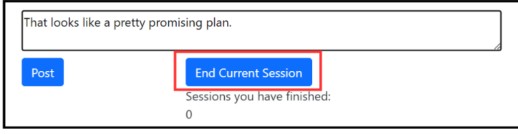

You will need to have at least **20** utterances (together with the other speaker) until the "End Current Session" button becomes enabled.

We recommend all participants to end the session as natural as possible.

Participants who provide high quality dialogue data will be rewarded.

By clicking the **End Current Session** button, you will move the time forward with a period of time (e.g., 1 week, 1 month, etc.) and start a new session.

Figure 8: The instruction we use for data collection.

## Step 5: Confirm the progress and new events

> 1 month has/have passed. During this time, the following events happened.
> **Finished Progress:**
> Think about the theme of the album.
> **Life Events:**
> You learn how to speak Spanish.
> You launch a buisness and start a new venture.
> You have a wedding anniversary.
> **World Events:**
> Argentina won the championship of World Cup.
> Fatal winter storm leaves millions without power in North America with -45C temperatures.
> Retirement: New Rules Are Coming For 401(k) and IRA Accounts.
> **Future Plans:**
> Begin writing and recording songs.
> Select the 10 best songs for the album.
> Begin producing the tracks.
> You are planning to meet with your team on Monday to discuss the progress of your project
> You are planning to take Friday off and spend the day with your family and friends.

A new session will provide you the following information:
**Progress** describes the progress of your initial event.
**Life events** are some events you could have done during that time (something happen to you).
**World events** are some events happen in the world (news).
**Future plans** are something you will do from now.

Select what events to talk about based on your social skills in each session.
You do NOT have to include every event.
Instead, make the conversation as **natural** as possible while using these events smartly.

Repeat step 3-5 until the end of the multi-session conversation.
You will need to finish at least **4 sessions** to complete this HIT.

# Evaluation Criteria (Important)

1. **Talk about the events/progress.** You need to talk about the events, rather than having free chitchats.
2. **Naturalness.** You are able to do free conversation, however, you are responsible to make your conversation natural.
3. **Fluent English.** Your must communicate in English fluently.
4. **Non-offensive.** You are not allowed to post any offensive, abusive and harmful messages in the conversation.

# Examples

## What we want:

**Ask progress of appropriate events:**
Session 1: ... talked about thesis writing and watching a movie ...
========== Time gap: 3 months ===========
Session 2:
A: Hi, it's been quite a long time, how's everything?
B: I'm doing great, how about you?
A: Great, thanks for asking. How is your thesis writing? Any progress?
**Don't ask about the movie, because it's been 3 months**

## What we do NOT want:

**1. Talk about the HIT it self:**
A: What's your topic?
B: My topic is about having a vocation.
**2. Not natural:**
A: Hi, how are you?
B: Hi, I'll have a trip.
A: I'm doing great.
**3. Filler utterances:**
A: Hi.
B: Hi.
A: Hi.
B: Hi.
**4. Not talking about the event:**
A: Hey, do you have a boyfriend?
OR
A: What is your name?

Figure 9: The instruction we use for data collection.

You are matched with your task partner. You can start the conversation by typing in the message area and hit the Post button.

Chat history

Your partner : And I really enjoy the processes. I think that's a great experience.

You : Glad to hear that, good luck with the house building.

Your partner : Thanks, you too. Good luck with your campaign.

You : Thanks, nice talking with you, see you soon.

Your partner : See you.

(The above conversation was 1 year ago)

Your message ...

Post      End Current Session

Sessions you have
finished: 1

1 year has/have passed. During this time, the following events happened.
**Finished Progress:**
You finished the previous progress and started the following new event.
You just started to write a new business plan, which would take about 2 weeks.
**Life Events:**
You get divorced.
You buy a new house.
You finish the training process of your new company.
**World Events:**
Nintendo Slashes Prices Of Switch eShop Games In New Year Sale (North America)
India makes negative COVID test mandatory for five countries.
November Carbon Emissions In Europe Lowest In 30 Years.
**Future Plans:**
You are planning to spend time volunteering at a local charity organization once a week throughout the month.
You are planning to attend a professional conference in the middle of the month to learn about the latest industry developments.

Figure 10: The interface for speakers to have conversations.

In the following conversation, the speakers are engaging in some events that take a certain amount of time. Extract such events and estimate the expected time to finish these events.

*Instances ∗N*

**Conversation:**
A: Hi how are you?
B: Yes I am fine and how are you doing today?
A: Doing good. What is the plan for tonight?
B: Not yet planed for something. I just started with preparing and executing a social media marketing campaign for my company.
A: Oh are you busy in that?
**Events:**
B: executing a social media marketing (about 3 months)

*Question*

A: Hi, how are you doing?
B: I'm doing great, how about you?
A: I'm also doing good. I'm just busy with my paper writing as the deadline is approaching.
**Events:**
<extracted events>

Table 12: An example of few-shot prompting with information explaining

*Instances ∗N*

**Conversation:**
A: Hi how are you?
B: Yes I am fine and how are you doing today?
A: Doing good. What is the plan for tonight?
B: Not yet planed for something. I just started with preparing and executing a social media marketing campaign for my company.
A: Oh are you busy in that?
**Events:**
In the above conversation, speakers talked about the events they are engaging. A is engaging in **something is not mentioned**. B is engaging in **executing a social media marketing, which takes about 3 months.**

*Question*

A: Hi, how are you doing?
B: I'm doing great, how about you?
A: I'm also doing good. I'm just busy with my paper writing as the deadline is approaching.
**Events:**
In the above conversation, speakers talked about the events they are engaging. _____ is engaging in _____. _____ is engaging in _____.

Table 13: An example of few-shot prompting for slot filling, where the LLM is required to fill in the blanks according to given instances.

| Instances $*N$ |
| --- |
| **Conversation:** |

A: Hi how are you?
B: Yes I am fine and how are you doing today?
A: Doing good. What is the plan for tonight?
B: Not yet planed for something. I just started with preparing and executing a social media marketing campaign for my company.
A: Oh are you busy in that?

*Questions*

**Question 1:**
Did speaker A mention any events that speaker A is engaging? Answer with Yes or No
Answer: _____ (No)

**Question 2:**
Did speaker B mention any events that speaker B is engaging? Answer with Yes or No
Answer: _____ (Yes)

**Question 3:**
What are the events that speaker B is engaging? Answer the the content of the event and an estimated time to finish that event.
Answer: Speaker B is engaging in executing a social media marketing, which takes about 3 months to finish.

**Question 4:**
Give a rough schedule of the events that B is engaging within the estimated time.
Answer:
Week 1-4: Research the target audience and their social media habits. Identify the platforms that the target audience uses most.
Week 5: Create high-quality, engaging content that aligns with the campaign goals and appeals to the target audience.
...

Table 14: An example of question answering style prompt to extract the events and their duration.

Given an event and a rough duration to finish this event, generate a short schedule for finishing this event. If it requires more information to get the schedule, roughly estimate one.
Generate the answer with the following requirements.
#requirements:
1. The generated schedule should be finished within the duration of the event.
2. The format should be the same as the Answer shown in the #Example.
3. Each schedule has at least 7 steps.

**#Example:**
event: getting driving license
duration: 2 months
Answer: one week for learning rules, 2 weeks for practicing, 2 weeks for passing exams, one week for road check, one week for getting license
**# Question:**
event: {...}

Figure 11: The prompt we use to generate the steps towards finishing a life event.

Given a list of events, generate a short schedule for finishing each event in JSON format. If it requires more information to get the schedule, roughly estimate one.
Generate the answer with the following requirements.
#requirements:
1. Must be a valid json file that can be parsed by python json package. Pay attention to the commas.
2. The format should be the same as the Answer shown in the #Example.
3. Each field in the Answer is a list.

**#Example:**
events: {"speaker_1": ["just started a one-year collage program."],"speaker_2": ["getting driving license"]}
Answer: {"speaker_1": ["1 month for initiating, 2 months for basic courses, 3 months for main courses, 2 months for selecting thesis topics, 2 months for finishing thesis, 1 month for preparing defense."],"speaker_2": ["one week for learning rules, 2 weeks for practicing, 2 weeks for passing exams, one week for road check, one week for getting license"]}
**# Question:**
List of events:
{ "speaker_1": ... }

Figure 12: The prompt we use to get schedules for the pre-defined events during training and extracted events during inference.

**First session:**
You are having a multi-session conversation your friend with the following conditions and example.
# conditions:
1. You are updating your current daily events.
2. You are aware of the rough time estimation to finish different events.

# your events:
{1. You are currently working on a research project, which takes about 3 months.
2. You are finding a new apartment, which takes about 3 weeks.
3. You are planning to have a short vocation to the beach side during this weekend.
...}

**Subsequent session:**
It's been {two weeks} and your friend is having another conversation with you. Continue the conversation with the following conditions.
# conditions:
1. You are updating your current daily events.
2. You are aware of the rough time estimation to finish different events.

# your events:
{1. You started taking an online course to learn programming, which takes around 3 months.
2. You are planning to watch a movie which takes around 3 hours.}

Figure 13: Prompts for generating conversations from ChatGPT (with gap). Content in { } is replaced with same events and time gaps as in the collection process of other RAG models.

## E   Prompt for collecting conversations from ChatGPT

Figure 13 is the prompt we use for collecting conversations from ChatGPT. We use two prompts to generate the first session and subsequent sessions. For the first session, we prompt ChatGPT to have a multi-session conversation with the same event settings as in other TA-RAG models. This prompt contains the conditions and events information. After generating the first session, we use a different prompt for generating the subsequent sessions. In the time-aware case of this prompt, we explicitly tell ChatGPT that there is a time gap between this session and the previous session and previous dialogue history is included as part of the prompt. First several utterances will be removed when the conversation is longer than the maximum input of ChatGPT. When generating time-aware ChatGPT conversations, we add another section of progress labels or schedules into the prompt we use for generating subsequent sessions.

## F   Human Evaluation Questionnaire

We ask the annotators in total 11 questions (Table 15) and ask them to choose one system over another one. For question Q3, Q6, Q9 and Q11 we ask annotators to provide justifications to explain

the reason of their choices. The justifications are used for both filtering out invalid answers and analyzing users feedback. We consider the following two types of annotations as invalid.

First, we remove those annotations that are with an extremely short working time. Given the length of the dialogues and questions, we consider it impossible to finish the annotation within 200 seconds. Therefore, any annotations that are with a work duration less than 200 seconds are directly filtered out. Second, we remove those annotations that are with unreasonable justifications. For instance annotators may give ["good", "good", "good", "good"] as justifications, which are obvious signs of the worker not engaging in the task. We perform pilot tests to select good annotators for our tasks. The annotators are required to have finished more than 2000 tasks, with an approval rate over 97% and be fluent in English.

## G   Sample Conversations

Table 16~18 show an example of the conversations generated by TA-RAG (both) (Speaker B).

| Questions |
| --- |

*Naturalness*

Q1. Which dialogue do you think is more natural like two friends updating their daily life?
Q2. Which dialogue do you think is more like a dialogue between normal friends?
Q3. In which dialogue do you think the speaker B talks more naturally?

*Informativeness*

Q4. Which dialogue do you think provide more information about the speakers' daily events?
Q5. In which dialogue do you think the speaker B asks annoying questions more
frequently about events that have no significant progress?
Q6. In which dialogue do you think the speaker B cares more about speaker A's daily events?

*Relevance*

Q7. Does the speaker asks/talks about relevant events?
Q8. Which dialogue do you think sticks to the topics of updating events?
Q9. In which dialogue do you think the speaker chooses natural, relevant events to talk about?

*Time-awareness*

Q10. In which dialogue do you think speaker B can identify a time gap?
Q11. In which dialogue do you think speaker B is aware of the progress of relevant events?

Table 15: The questions we ask annotators for the human evaluation.

Events:
A: You watch a movie.
You are going to visit your grandparents.
B: You plan to go to a music festival.

A: I am doing well. I love the outdoors as well. What beach did you go to?
B: do you have any plans for the upcoming weekend? I'm planning to go to a festival.
A: That sounds like fun. I am going to a concert this weekend. What kind of music do you listen to?
B: What about you? What do you do for fun? I like to dance and go to concerts.
A: I like a lot of different kinds of music. My favorite band is Led Zeppelin.
B: what is your favorite thing to do in the evening? I love to go dancing with my friends.
A: I like going out to eat with my family. We like to eat out a lot. Do you have a favorite food?
B: That's great. I also like to go out with friends. I like dancing and dancing.
A: Dancing is a great way to get out and have fun. What type of dancing do you enjoy?
B: I love dancing, but I don't dance much. I'm more of an outdoor person.
A: Outdoor dancing is great too. It is so much fun to be out in nature.
B: I love that movie. It's one of my all time favorites. Have you seen it?
A: No, I haven't seen it. I have heard it is good though. What is it about?
B: I love watching movies and listening to music. What's your favorite genre of music to dance to?
A: I love pop music. I listen to it all the time while I work out at the gym.
B: That's nice. What about your family? Are you planning to do anything special for the holidays?
A: My family is great. We are going to visit my grandparents. They live in Florida.
B: Do you like to watch movies or tv shows? I am a big fan of the genre.
A: I like movies and tv shows too. What are some of your favorite shows to watch on tv?
B: I'm going to watch a movie with my brothers and sisters. It will be fun.
A: What movie are you going to see with your siblings? I hope it's a good movie.

Table 16: Session 1 of an conversation generated by TA-RAG (both).

| |
|---|
| Gap: 1 week |
| Events: |
| A: You watch tv dramas. |
| B: You plan to go hiking. |

B: what about you what are your plans for this weekend? What are you up to? I plan to go hiking.
A: Hiking sounds like a good idea. I might go for a hike too. I've never been hiking before.
B: how were your grandparents? are they doing great?
A: Thanks for asking, they are doing great. And I had a great time with them. We watched Game of Thrones.
B: that's great, I'm glad to hear that. How about you, what's going on?
A: I just finished watching the last season of House of Cards. It was a great show.
B: yes, I like it too, it is a romantic comedy and romance comedy.
A: It really is. I can't wait for the new season to come out in 2019.
B: do you know any good romance movies? I really like romance and thrillers movies.
A: I do! I love The Notebook. I think it's my favorite movie of all time.
B: I love romantic movies, especially romantic ones. I just watched a movie about romance.
A: That sounds interesting. I'll have to check it out. What did you think of the movie?
B: I like romantic drama and thriller movies, do you also like those genres of movies?
A: I do like those types of movies. I especially like the thrill of a good thriller.
B: yes, i love romance, action, adventure, thriller, mystery, suspense.
A: Have you ever seen the movie "The Sixth Sense" by David Suchet?

Table 17: Session 2 of an conversation generated by TA-RAG (both).

| |
|---|
| Gap: 2 months |
| Events: |
| A: You prepare for the coming tests. |
| B: You just started to learn to play piano, which takes about 5 months. |

B: I'm doing great, thanks for asking. How is your day going? What have you been up to lately?
A: My day is going well, thank you! I've been studying for a test I have tomorrow.
B: what are you planing to do for the weekend? I am planning to see a movie.
A: I plan on going to the movies as well! I'm looking forward to seeing the new Star Wars movie!
B: do you want to go with me? I can take you to the theater to see the movie.
A: That would be great! I'll bring my boyfriend as well, he's the only one I've ever gone with.
B: I like watching movies, tv shows, documentaries, documentarys, tv series and documentaries.
A: I love documentaries! I watch them every night before I go to bed. Whats your favorite?
B: do you have any other hobbies or hobbies? I just started learning how to play the piano.
A: That's so cool! I play the drums. I started when I was a teenager.
B: I'm planning to take a trip to the beach in the next few months. I hope you can join me.
A: I would love to! I have never been to a beach before. Where will you be going?
B: It is a place somewhere near to the south east coast.
A: That sounds like a lot of fun. I hope I could go together with you.

Table 18: Session 3 of an conversation generated by TA-RAG (both).

# H TA-RAG model's performance over different session gaps

Table 19 shows the numbers of different session gaps in evaluation experiments. Figure 14 and 15 are the performance of TA-RAG models with different time-aware information over different session gaps. Although we are able to observe more stable naturalness for TA-RAG (both) model, that does not hold in TA-RAG (schedule). Informativeness is also more stable in TA-RAG (progress), however shows larger deviation for TA-RAG (both) and TA-RAG (schedule).

| Session gap | #sessions |
|-------------|-----------|
| minutes     | 82        |
| hours       | 90        |
| days        | 65        |
| weeks       | 78        |
| months      | 62        |
| year        | 73        |

Table 19: Number of different session gaps in our experiments.

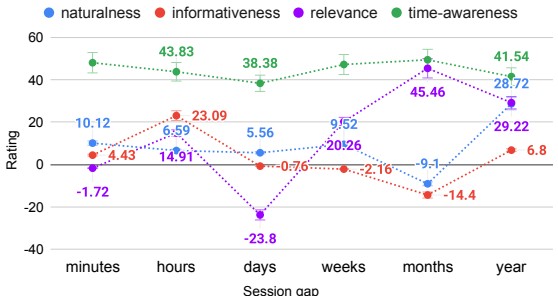

Figure 14: Performance of TA-RAG (progress) over different session gaps.

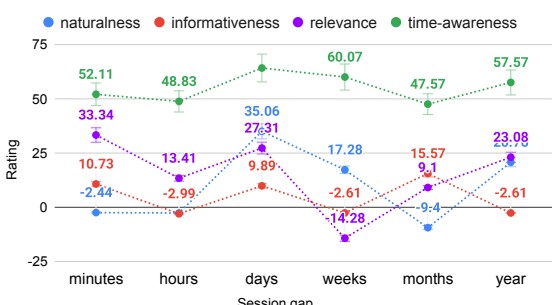

Figure 15: Performance of TA-RAG (schedule) over different session gaps.