# OpenReview forum: "Mind the Gap Between Conversations for Improved Long-Term Dialogue Generation"
_EMNLP/2023/Conference — EMNLP 2023 Findings_

### Official Review · Reviewer_hTeo · 2023-08-04

**Soundness:** 3

**Excitement:**

4: Strong: This paper deepens the understanding of some phenomenon or lowers the barriers to an existing research direction.

**Paper Topic And Main Contributions:**

The authors of this paper introduce a new dataset, called GapChat, of multi-session dialogues. This way, they want to investigate how generation models can be made time-aware. They extend the existing MSC dataset by (1) looking at long-term gaps (instead of short) and (2) paying attention to the relation between topic events and the duration of gaps.

**Questions For The Authors:**

- Section 3.1 (p.3): Were past real-world event also ordered chronologically as they happened in the real world? Or are they shuffled randomly as the life events?

- l.230-1: "$7 per hour on average in accordance with 231 living wage standards" -> How do you know that these are standard living wages, if you recruited crowd-worked through Amazon Mechanical Turk and the only requirement was a fluency in English. Aren't living wage standard higher in many English speaking countries?

- l.244-6: "For all subsequent sessions, speakers are provided with updates on the progress of their ongoing events based on the schedules and the time gap." -> How did you decide on these updates? Randomly or manually?

- l.271-2: train/val/test split
(a) How did you split the data in train/val/test? Randomly or stratified across session length?
(b) Moreover, did several conversation about the same timeline occur? (Not specified in paper as far as I could see.) If so, did you take this into account while making the train/val/test split? Did all conversations on a specific timeline occur in only in 1 unique set (either train, val or test)?
(c) Is the test set actually used, as your models are evaluated by humans?

- l.285: Could you in your formulas distinguish between ground truths and estimates/generations close to this ground truth?

- l.306-7: "The prompt which resulted in the highest extraction performance(...)" -> How did you calculate this performance? What if spans didn't correspond? Was it calculated in an end-to-end or pipeline setup?

- l.349-51: "During training time, events and different types of time information (progress labels and schedules) are combined with dialogue history" -> What happened when this was too long to fit into the model? How did you truncate?

- l.353-5: "When generating a subsequent session, the gap information is also added as part of the context following the method in MSC." -> Could you provide some more details to clarify this sentence.

- Table 2 (p.6): What metric is it? What scale are these scores on?

- l.365-6: Could you provide more details on the fine-tuning here?

- l.373: The ChatGPT "with gap" model seems to also be a time-aware model. Although it doesn't have a lot of information about the times, the model does know between which instances there was/wan't a gap. Why did you decide to frame it as a time-unaware model?

- p.6: It is very difficult to judge this section, without having been given more details on the MSC-RAG model. How does this model work? Does it use prompt-based learning or is the input encoded in an embedding to fine-tune the model?

- l.437-448: By how many annotators was each instance annotated?

- l.447-8: How many annotators were involved in the Fleiss' Kappa study? How many instances did you use to calculate this score?

**Reasons To Accept:**

- The authors present a very interesting novel dataset that can become a valuable research for embodied and real-life conversational applications.

- The merit of this dataset is demonstrated through ML experiments. Systems that receive time-aware information perform generally better than their time-unaware counterparts.

**Reasons To Reject:**

- Although the paper starts very promising and there is definitely a lot of potential in the GapChat dataset, there are currently too many crucial details missing (on the dataset, implementation of models and results) to reproduce the results (see "Questions For The Authors").

- The formulas in the task definition seem to be wrongly formulated.

- In terms of modelling, there seems to be relatively little innovation. The authors give prompts to ChatGPT and use an existing model on their dataset. Nevertheless, it was difficult to properly judge this section as quite some implementation details were missing, so my judgement might be wrong here.

**Reproducibility:**

3: Could reproduce the results with some difficulty. The settings of parameters are underspecified or subjectively determined; the training/evaluation data are not widely available.

**Reviewer Confidence:**

2: Willing to defend my evaluation, but it is fairly likely that I missed some details, didn't understand some central points, or can't be sure about the novelty of the work.

**Typos Grammar Style And Presentation Improvements:**

- l.32: "may not be well-represented explicitly" -> may not be wel represented vs. may not be represented explicitly
- l.118: "coversation" -> conversation
- l.143-4: "to pay attention the relation" -> pay attention to
- l.271: "split the data to" -> split into
- l.280: the final x between {} should be x_n. Especially if you talk about all utterances as x_i
- l.285: Make a distinctions between ground truth (e.g., E) and estimation of ground truth (by circumflex, e.g., Ê)
- l.285: y = argmax P(y|H, progress, t) -> this formula does not make sense. See previous comment.
- l.329: "a the"
- l.449-460: Would it not be better to place these training details in the section on models instead of the section on evaluation?

---

> ### Author Rebuttal · Authors · 2023-08-29
>
> Thank you for your thorough review and many questions.  We would like to discuss them in two parts: (1) aspects of the procedure that may affect the presented results, (2) other aspects which are for more documentary purposes.
>
> ----------
> (1)
> ----------
> `Q:` How did you split the data in train/val/test? Did several conversation about the same timeline occur? Is the test set actually used?
>
> `A:` Regarding the train/valid/test split, we split the data randomly based on session length. Based on the sheer number of variables in constructing a timeline (choosing of the many events, the time gaps, how those events relate to world events, where in the conversation those events are introduced) it is extremely unlikely to have significant overlap between any two timelines.  Indeed, examining the dataset reveals that no timelines have identical event sequences, which is true both between train/test sets, but also within any set. We did not save certain events for the evaluation, and we acknowledge the concerns regarding the potential leakage of events from the training set to the test set. In our dataset, we found that 1.54\% of life events sequences and 2.31\% of world events sequences in the test set also appear in the training set, though they are presented as part of distinct timelines in the two sets.
>
> `Q:` More details on the MSC-RAG model? More details on the fine-tuning of RAG (FT)? What happened when this was too long to fit into the model? How did you truncate?
>
> `A:` The MSC-RAG model is a transformer model that stores past dialogue history as documents and employs a retrieval system to select certain history portions for inclusion in the final encoding. It has demonstrated effective performance in handling extensive context. Various methods for storing documents exist (such as storing them as DPR weights or using a FAISS index). We adopt the approach that yielded the best performance in the MSC paper [1] (the DPR approach) for all our RAG models.
>
> To obtain the RAG (FT) model, we fine-tune the MSC-RAG model on GapChat. The fine-tuning involves two steps:
> Calculate new DPR weights using GapChat available at \url{https://github.com/facebookresearch/DPR}.
> Continue training the MSC-RAG model with the new DPR weights using the ParlAI platform.
> Regarding the training settings, we also follow the approach from MSC. In our time-aware models, we concatenate different time-related information (progress labels, gap information, schedules) as raw text with the dialogue history in the encoding. If the dialogue history becomes excessively long, we truncate several utterances from the beginning of the conversation.
>
> `Q:` How many annotators was each instance annotated? How many annotators were involved in the Fleiss' Kappa study? How many instances did you use to calculate this score?
>
> `A:` As stated in Lines 440-441, during the evaluation process, we employed 10 annotators to annotate each instance, resulting in a total of 66 annotators. Fleiss' Kappa was calculated across 150 conversations (600 instances: $50\times3 + 50\times4 + 50\times5$) involving all 66 annotators.
>
> [1] Beyond Goldfish Memory: Long-Term Open-Domain Conversation (\url{https://aclanthology.org/2022.acl-long.356/})
>
>
> ----------
> (2)
> ----------
> Regarding other questions of a more documentary / reproducibility standpoint, we want to first reiterate that the dataset and all code used in its creation is available upon acceptance, and answers to these questions are observable in these resources.  In addition we will incorporate the answers to the following questions into the appendix or into the main content where appropriate:
>
> `Q:` Were past real-world event also ordered chronologically as they happened in the real world?
>
> `A:` Yes.
>
> `Q:` How did you decide on these updates of life events?
>
> `A:` We prompted ChatGPT to provide plausible multi-step decompositions for long-duration life events, and to estimate their durations.
>
> `Q:` Aren't living wage standard higher in many English speaking countries?
>
> `A:` We limit the locale of workers to North America.  We mistook minimum wage for living wage which is now estimated to be much higher in these countries (\$14-20), and we will correct this.  We strive to accommodate workers fairly, and offered \\$7 based on minimum wage and the average MTurk rate (\\$3), but unfortunately the grants supporting this work are not from these countries which makes it difficult to further increase the compensation per annotator.
>
> `Q:` How did you calculate this performance of the prompts? What if spans didn't correspond?
>
> `A:` Regarding the prompt selection, please refer to the response to reviewer j5Mt. We check the generated content with different prompts manually in a pipeline setup, in which for some prompt templates we send multiple prompts and the consequent prompt depend on the answers to previous prompts. We check both the content and format3  If the spans didn't correspond or the answers are not the expected format, they will be labelled as format errors.
>
> `Q:` Why did you decide to frame ChatGPT (with gap) as a time-unaware model?
>
> `A:` Thank you, it is a good point and we understand how it could be a point of confusion.  It is time-aware in the sense that it is given information about gaps and their durations.  The model does not receive event and event progress information, which is the distinction we were making with "time-aware", but we see how the terminology could be improved and clarified.
>
> `Q:` What metric is in Table 2? What scale are these scores on?
>
> `A:` Table 2 scores are from human evaluation using ACUTE-Eval, which is a preference based task. We display two sessions generated by two different models (one target model and one baseline model) and ask annotators to rate one model over another one. We calculate the human ratings to get the numbers in Table 2, where negative numbers indicate that annotators rate the target models as being worse than the baseline model and positive numbers indicate the opposite. In our experiments, we use RAG (FT) model as the common baseline and compare all other models against it to get Table 2. The scores are calculated in percentages. For instance $13.75$ means for all evaluation instances, $13.75$ more annotators voted for the target model being better than RAG (FT) on the specific attribute.
>
> `Q:` Typos in the text and formula?
>
> `A:` Thank you for pointing out some minor typos.  We will perform further editing and seek proofreading by native speakers for a final draft. And regarding the writing style in the formula, we will represent predicted/generated results as
> $ \hat{y} = \\{ \hat{y_1},... \hat{y_i} \\}$ to distinguish from ground truth. That makes the formula:
>
> $$ p\(\hat{y_i}|H, progress, t \) =  \prod P(\hat{y_i}|\hat{y_1},...,\hat{y_{i-1}, H, progress, t})$$
> , where $H$ represents the dialogue history $\{x_1,...x_{i}\}$, $progress$ is the corresponding progress of the events and $t$ is the time gap.

---

### Official Review · Reviewer_b1Jg · 2023-08-11

**Soundness:** 4

**Excitement:**

4: Strong: This paper deepens the understanding of some phenomenon or lowers the barriers to an existing research direction.

**Paper Topic And Main Contributions:**

This paper addresses the problem of multi-session chat (MSC), where a conversational agent needs to maintain coherent and engaging dialogues with a user across multiple sessions with time gaps. Previous MSC datasets have short time gaps and do not consider the impact of time on topic transitions and information updates. To fill this gap, the paper proposes GapChat, a novel MSC dataset that incorporates time-aware information and simulates realistic scenarios of MSC. The paper also proposes a time-aware dialogue model, TA-RAG, that leverages GapChat to generate more appropriate and informative responses. The paper evaluates the proposed dataset and model using both automatic and human metrics, and shows that they outperform existing baselines.


**Reasons To Accept:**

1. The paper introduces a novel and challenging task of time-aware MSC, which has not been well-studied before.
2. The paper presents a reasonable and effective pipeline to construct GapChat, a high-quality MSC dataset with time-aware information.
3. The paper proposes TA-RAG, a time-aware dialogue model that exploits GapChat to generate more relevant and informative responses.
4. The paper conducts extensive experiments to demonstrate the superiority of GapChat and TA-RAG over existing MSC datasets and models.

**Reasons To Reject:**

1. The human evaluation may be subjective and influenced by various factors. It would be desirable to have some automatic metrics that can measure the time-awareness of the dialogue model.




**Reproducibility:**

5: Could easily reproduce the results.

**Reviewer Confidence:**

3: Pretty sure, but there's a chance I missed something. Although I have a good feel for this area in general, I did not carefully check the paper's details, e.g., the math, experimental design, or novelty.

---

> ### Author Rebuttal · Authors · 2023-08-29
>
> Thank you for your reviews and your appreciation of the strengths of this paper.  We agree that an automatic evaluation would be a great inclusion in the paper, but unfortunately we had difficulty finding a metric which can automatically capture time-awareness in a way that is reasonably accurate.  We believe the best automatic metric would itself use GPT, and since GPT is used in the identification of events and durations used in generation, we fear this would likely result in something that is far more biased and preferential to our models specifically.  We are certainly open to suggestions.  But in that sense we believe the human evaluation is more fair and more reflective of overall quality and time awareness.  In our human evaluations, we did control for a variety of priming, order, and presentation biases. We performed pilot studies to select trustworthy annotators and we had 10 annotators for each question.

---

### Official Review · Reviewer_j5Mt · 2023-08-12

**Soundness:** 4

**Excitement:**

4: Strong: This paper deepens the understanding of some phenomenon or lowers the barriers to an existing research direction.

**Paper Topic And Main Contributions:**

This paper presents a novel multi-session dialogue dataset called GapChat, which incorporates time-related information, such as the intervals between sessions and the progression of speaker-related events. To emulate real-life situations, the paper distinguishes between life events and world events. Additionally, the paper introduces a method for integrating temporal modeling in models utilizing the GapChat dataset.

**Reasons To Accept:**

1. This paper provides a fresh perspective on multi-session conversation research.


2. The process behind the construction of the GapChat dataset is meticulously detailed and explained.


3. The approach for integrating temporal information into multi-session conversation models is both well-defined and interesting.


4. The results from the human evaluation substantiate the authors' claims.

**Reasons To Reject:**

1. I noticed a minor typo in line 329: "a the progress...". While I did not spot other errors, I recommend the authors conduct thorough proofreading as there may be additional typos.


2. While ChatGPT is an impressive model, the current state of this paper seems heavily reliant on ChatGPT's accuracy in event extraction and duration estimation. Did the authors undertake empirical studies specifically related to prompt tuning?

**Reproducibility:**

5: Could easily reproduce the results.

**Reviewer Confidence:**

4: Quite sure. I tried to check the important points carefully. It's unlikely, though conceivable, that I missed something that should affect my ratings.

---

> ### Author Rebuttal · Authors · 2023-08-29
>
> Thank you for your reviews and your appreciation of the strengths of this paper.  Regarding the typos, thank you for pointing them out.  We will perform further editing and seek proofreading by native speakers for a final draft.
>
> Regarding the prompts, we did explore many prompt strategies in support of this research, and reported what we believe to be the most important design decisions.  These are presented in Appendix D.2, with Table 11-13 showing the different types of prompt formats explored. To be more specific, we extracted 51 conversations from our dataset and manually extracted the events as gold labels. We then evaluated the accuracy of a total of four different types of prompts (indicated as `instruction only`, `slot filling`, `conditions`, and `QA`). We assessed: 1) whether the response contains correct event information, and 2) whether the response is in the specified format, as given in the prompt. The final prompt we chose for our paper (`conditions`) achieved the highest accuracy of 76.19 for event information and 100.00 for format. The complete results are as follows:\
> For content:\
> Instruction only: 38.10, Slot filling: 71.43, Conditions: 76.19, QA: 74.41\
> For format:\
> Instruction only: 95.24, Slot filling: 71.43, Conditions: 100.00, QA: 96.13

---

### Meta-Review · Area_Chair_TQza · 2023-09-19

**Recommendation:** 3

**Metareview:**

This paper presents a multi-session chat dataset where each session is separated by a time interval. During this time interval, speaker-related or world-related events may change. The authors also propose a model to handle such dialogues. The reviewers appreciate the novelty of this dataset as well as its thorough presentation and evaluation but also raise some concerns, including confusion in some formulas, lack of novelty of the proposed model (not the dataset), as well as some issues with the evaluation. The authors address the main concerns and therefore I recommend accepting this paper.

---

### Decision · Program_Chairs · 2023-10-07

**Decision:**

Accept-Findings

**Comment:**

This paper presents a multi-session chat dataset where each session is separated by a time interval. During this time interval, speaker-related or world-related events may change. The authors also propose a model to handle such dialogues. The reviewers appreciate the novelty of this dataset as well as its thorough presentation and evaluation but also raise some concerns, including confusion in some formulas, lack of novelty of the proposed model (not the dataset), as well as some issues with the evaluation. The authors address the main concerns and therefore I recommend accepting this paper.